# Chitosan Hydrogel Supplemented with Metformin Promotes Neuron–like Cell Differentiation of Gingival Mesenchymal Stem Cells

**DOI:** 10.3390/ijms23063276

**Published:** 2022-03-18

**Authors:** Shanglin Cai, Tong Lei, Wangyu Bi, Shutao Sun, Shiwen Deng, Xiaoshuang Zhang, Yanjie Yang, Zhuangzhuang Xiao, Hongwu Du

**Affiliations:** 1School of Chemistry and Biological Engineering, University of Science and Technology Beijing, Beijing 100083, China; 13578879366@163.com (S.C.); b20200391@xs.ustb.edu.cn (T.L.); s20190900@xs.ustb.edu.cn (W.B.); 41822066@xs.ustb.edu.cn (S.D.); b20190373@xs.ustb.edu.cn (X.Z.); s20190918@xs.ustb.edu.cn (Y.Y.); g20198933@xs.ustb.edu.cn (Z.X.); 2Daxing Research Institute, University of Science and Technology Beijing, Beijing 100083, China; 3Institutional Center for Shared Technologies and Facilities, Institute of Microbiology, Chinese Academy of Sciences, Beijing 100101, China; shutaosun@im.ac.cn

**Keywords:** chitosan hydrogel, metformin, gingival mesenchymal stem cell, neuronal differentiation

## Abstract

Human gingival mesenchymal stem cells (GMSCs) are derived from migratory neural crest stem cells and have the potential to differentiate into neurons. Metformin can inhibit stem–cell aging and promotes the regeneration and development of neurons. In this study, we investigated the potential of metformin as an enhancer on neuronal differentiation of GMSCs in the growth environment of chitosan hydrogel. The crosslinked chitosan/β–glycerophosphate hydrogel can form a perforated microporous structure that is suitable for cell growth and channels to transport water and macromolecules. GMSCs have powerful osteogenic, adipogenic and chondrogenic abilities in the induction medium supplemented with metformin. After induction in an induction medium supplemented with metformin, Western blot and immunofluorescence results showed that GMSCs differentiated into neuron–like cells with a significantly enhanced expression of neuro–related markers, including Nestin (NES) and β–Tubulin (TUJ1). Proteomics was used to construct protein profiles in neural differentiation, and the results showed that chitosan hydrogels containing metformin promoted the upregulation of neural regeneration–related proteins, including ATP5F1, ATP5J, NADH dehydrogenase (ubiquinone) Fe–S protein 3 (NDUFS3), and Glutamate Dehydrogenase 1 (GLUD1). Our results help to promote the clinical application of stem–cell neural regeneration.

## 1. Introduction

The repair of central nervous system disorders plagues the medical community, including traumatic brain injury [1]; spinal cord injury [2]; and neurodegenerative diseases, such as Parkinson’s disease, Alzheimer’s disease, etc. [3]. The main source is the lesions of the nerve cells, for which the main symptoms are the loss of neurons and the imbalance of glial cells. Studies have shown that there are still new nerve cells in the adult brain, while the content of new nerve cells in neurodegenerative diseases is less. Lacking cytogenesis of nerve cells may be the reason why various treatment therapies fail for neurodegenerative diseases. Exogenous supplementation of neural stem cells is considered to have broad prospects for neurodegenerative diseases [4]. Neural stem cells have womeithe potential to differentiate into neurons and glial cells. They also perform self–renewal and repair damaged brain tissue through paracrine function and the replenishment of nerve cells [5].

However, the established neural stem cells are mostly derived from mice, so their clinical applications are limited. The source of human–derived neural stem cells is insufficient, and they are prone to tumor formation in the body [6]. There is an urgent need for a neural stem cell with a wide range of sources, as well as strong neural stemness.

The dental–tissue–derived mesenchymal stem cells originate from the neural crest during embryonic development, and they are the precursor of nerve and bone tissue [7]. The ability to differentiate into nerve–like cells and the potential for nerve regeneration from the tooth and oral stem cells has been demonstrated in vivo [8]. Moreover, undifferentiated odontogenic stem cells proved to be neuroprotective when they were transplanted into animal models of nerve injury [9]. Because only a small fraction of the transplanted stem cells can differentiate into neural lineages in vivo, the clinical efficacy of transplantation is limited. In addition, there is a risk that undifferentiated stem cells may produce an undesirable lineage at the site of transplantation, which can impede nerve regeneration and is known as fibrotic scar tissue [10]. Pre–differentiation of stem cells into a neural lineage in vitro allows for the the expression of neuron–associated surface receptors and adhesion molecules before transplantation in vivo, thus accelerating the integration with the host nervous system in order to improve the clinical efficacy of transplantation treatment. GMSCs are the mesenchymal stem cells derived from gingival tissue, which can express neural–related markers, possesses multifunctional differentiation potential, and possess strong immune regulation ability [7]. So far, there is no report on the multistage method of GMSCs neural differentiation. El–bialy et al. [11] proved that low–intensity pulsed ultrasound (LIPUS) can enhance the neurogenic differentiation of GMSCs cultured in α–MEM supplemented with basic fibroblast growth factor (bFGF), insulin, DMSO, butylhydroxyanisole (BHA), KCl, valproic acid, and forskolin. Hsu et al. [12] used Neurobasal A medium and B27 supplement, in addition, Hsu et al. used a 0.1% gelatin–coating as a culture substrate. Recently, 3D–collagen hydrogel drives the conversion of GMSCs into NCSC/SCP–like cells [13]. Metformin is a drug commonly used to treat diabetes, but recently, its role in neurodevelopment has also been confirmed. The number of new neurons in mice with neurological damage increased after taking metformin [14]. Growth factors, such as (Epidermal Growth Factor) EGF, bFGF [15], and brain–derived neurotrophic factor (BDNF) [16], can effect cell growth, development, and differentiation, and they can regulate pathways by activating the receptors on the cell surface. Growth factors are often expressed in nerve tissues, can induce differentiation of neural stem cells and promote proliferation and development, and have neuroprotective effects. However, the growth environment of cells cultured in a petri dish is different from the growth environment in vivo, and there is less interaction between cells. Studies have shown that the three–dimensional (3D) cell culture system that simulates the growth environment of cells in vivo can enhance the differentiation potential of stem cells [17]. The scaffold for cell growth can not only change the biomechanical properties of the cells, allow the cells to grow into spheroids, and enhance the interaction between the cells, but also provide a suitable drug delivery microenvironment [18]. The chitosan hydrogel has received wide attention due to its biocompatibility and degradability. As a natural macromolecular carbohydrate, chitosan is similar to glycoprotein and is believed to be able to imitate extracellular matrix behavior in terms of structure and mechanical properties [19]. Therefore, in cell regeneration and differentiation culture, chitosan hydrogel can provide a unique cell scaffold. This study aims to use chitosan hydrogel containing metformin and growth factors. In this study, we used chitosan hydrogel as a culture scaffold and conducted segmental induction through growth factors embedded in the hydrogel and metformin in the medium to provide GMSCs with the most suitable growth and neural differentiation conditions (Figure 1). This study paved the way for the verification of the neurogenic potential of metformin on GMSCs and the construction of biomimetic scaffolds for disease treatment.

## 2. Results

### 2.1. Synthesis and Characterization of CS/β–GP

To build a 3D culture environment for mesenchymal stem cells, we mixed β–glycerophosphate sodium solution and chitosan solution in an ice–water bath. The results showed that CS/β–GP was crosslinked to form a hydrogel at 37 °C. The test tube was placed at an angle of 45° to the ground, and the hydrogel at the bottom of the test tube was firmly on the bottom of the test tube without flowing (Figure 1a). The chitosan solution does not solidify at 37 °C. The test tube was placed at an angle of 45° to the ground, and the glue at the bottom of the test tube flowed (Figure 1b). The CS molecules were tightly crosslinked with water molecules in the CS solution (Figure 1c), and there were no micropores suitable for cell growth and channels to transport water and macromolecules. In the crosslinked CS/β–GP hydrogel, the addition of β–GP made the CS form a perforated microporous structure (Figure 1d). According to the statistics of ImageJ, the pore size is between 20 and 40 microns, which is conducive to the growth of cells and the delivery of water and macromolecular substances. The Fourier infrared spectrum of CS/β–GP hydrogel shows that 1047.38/1068.25 cm^−1^ is the asymmetric stretching vibration of the phosphate group, and 958.105 cm^−1^ is the symmetric stretching vibration absorption peak of PO^43–^. The stretching vibration absorption peak of C=O (1644.55, 1651.85 cm^−1^) and the deformation vibration absorption peak of –NH (1538.83, 1557.12 cm^−1^) are red–shifted, indicating that the amine group on the CS molecular chain forms with the OH and PO^43–^ of the β–GP, hydrogen bonds, and complexes. On the other hand, the similarity of the infrared spectra of CS and CS/β–GP indicates that there is no chemical bond between CS and β–GP (Figure 1e).

### 2.2. Characterization of GMSCs

To verify the pluripotency of the isolated GMSCs and the effect of adding metformin as the induction drug on the pluripotency of GMSCs, the induction medium without metformin was called the control group, and the induction medium supplemented with metformin was called the Met group. We tested the osteogenic, adipogenic, and chondrogenic differentiation potential of GMSCs. On the 21st day of adipogenic induction, intracellular lipid droplets were observed by using Oil Red O staining, which proved the presence of adipocytes after induction. In the medium supplemented with metformin, GMSCs were found to have strong adipogenic differentiation ability (Figure 2a).

After osteogenic induction, osteoblasts stained with Alizarin red showed that calcium nodules appear on the cell surface, showing GMSCs can be differentiated into osteocytes. The Met group also had calcium nodules appear, indicating that GMSCs have strong adipogenic differentiation ability in the differentiation medium supplemented with metformin (Figure 2b).

After 21 days of culturing of chondrogenic induction medium, the chondrocytes were sliced and stained with Alcian blue. The chondrocytes slices were stained blue, and the chondrocytes induced by the induction medium supplemented with metformin could also be colored by Alcian blue (Figure 2c). In addition, our analysis of cell–surface biomarkers by flow cytometry showed that GMSCs cultured in complete medium (Figure 2d) and GMSCs cultured in complete medium supplemented with metformin (Figure 2e) are both highly expressed. Mesenchymal cell markers such as CD73, CD90, and CD105 were positive, but peripheral blood mononuclear cell marker HLA–DR, white–blood–cell surface marker CD19, and hematopoietic stem–cell markers such as CD45, CD11b, CD34 were negative. The percentages of cells expressing various biomarkers are shown in Table 1. The above data show that GMSCs have the characteristics of mesenchymal stem cells and the addition of metformin does not affect the mesenchymal stem cell characteristics of GMSCs.

### 2.3. Assessment of Cell Viability on Hydrogels

To detect the influence of CS/β–GP hydrogel on cell viability and to determine the optimal concentration ratio of cultured cells, we chose three proportioning schemes and mixed GFP–GMSCs with different concentrations of the hydrogel. When the volume ratio of β–GP to CS was 1:3 and 1:5, the fluorescence brightness gradually decreased between 0 and 24 h (Figure 3a,c). The quantified fluorescence intensity graph showed that the fluorescence intensity was greatly reduced between 0 and 24 h (Figure 3b,d), indicating that the cell viability of the cells under these two concentrations was reduced and cell apoptosis occurred. When the volume ratio of β–GP to CS was 1:9, the fluorescence brightness gradually increased between 0 and 24 h, and at 24 h, and the cells adhered and grew in the hydrogel and spread out in a fibrous shape (Figure 3e). The intensity quantification chart showed that the fluorescence intensity gradually increased between 0 and 24 h (Figure 3f), and the cell activity was the highest at this concentration.

### 2.4. Evaluation of the Differentiation of GMSCs into Neuronal Cells on the Hydrogel

It was demonstrated that when β–GP:CS was 1:9, the growth of cells was best, and chitosan hydrogel could provide adhesion sites for cells [20]. GMSCs can adhere to growth, and the 3D growth environment provided a higher level of proliferation and differentiation for cells than 2D culture. It was reported that metformin can promote the regeneration and development of neurons [21]. To verify the differentiation–promoting ability of metformin and to utilize the advantages of hydrogel in culture, we chose metformin as an inducing drug to induce GMSCs in a 3D growth environment to differentiate into nerve cells. In addition, growth factors, including EGF, bFGF, BDNF, and vascular endothelial growth factor (VEGF), which promoted differentiation and provided protection for neural stem cells, were added as cofactors to promote differentiation.

We assumed that the neural differentiation potential of GMSCs in Met–CS was higher than that in CS. To verify this conclusion, after 14 days of cell culture in chitosan hydrogel, the expression of neural–related markers at the level of RNA and protein was detected. Immunofluorescence detected the NES and TUJ1 proteins, which are neural–related markers, among which NES protein is a marker of neural stem cells, and TUJ1 protein is a marker of early immature neurons [22]. GMSCs derived from the mesoderm neural crest can express weak NES and TUJ1 proteins. Under the CS and Met–CS culture environment, the fluorescence intensity of NES and TUJ1 proteins was enhanced (Figure 4a,b), and this implied that the cultural environment of the hydrogel promoted the neural differentiation of GMSCs cells. Moreover, the quantitative graph of the average fluorescence intensity shows that the protein expression level of NES and TUJ1 in the Met–CS group had a significant increase (*p* < 0.001) compared with the CS culture system (Figure 4c,d). The mean fluorescence intensity was calculated by randomly selecting five fluorescence images from each well after 14 days. Consistent with the results of immunofluorescence, *NES* and *TUBB3* genes encoding NES and TUJ1 were observed to be highly expressed in the Met–CS culture system compared with the Con and CS groups (Figure 5a). In addition, in the Met–CS group and the CS group, *SOX1* has a significant increase in RNA level; Sox1 is believed to maintain the cell cycle and promote the self–renewal of neural stem cells [23]. In the Met–CS group, the expression of PAX6 gene increased, and PAX6 gene was expressed in the early neuroectoderm, which directed regulation of the differentiation of neural stem cells and was related to the growth and development of the optic nerve [24]. There was no significant change in the expression of MAP–2 and GFAP. MAP–2 is a neuronal phosphoprotein, which is related to neuron morphology and cytoskeletal dynamics and is considered a marker of mature neurons [25]. GFAP protein is glial fibrillary acid, which is considered a marker of astrocyte and oligodendrocyte [26]. Moreover, the semi–quantitative results of protein expression by Western blot also showed that NES and TUJ1 had a significant increase (*p* < 0.01) in expression in the Met–CS culture system compared with the control group and the CS group (Figure 5b). The statistical results showed that the CS culture system had a significant increase in the protein and gene expression of NES when compared with the control group (*p* < 0.01), while the increase in TUJ1 was not significant (Figure 5b). In RNA and protein expression, the expression of GFAP did not change significantly. It can be seen that GMSCs showed no significant change in the expression of markers in mature neurons after induction by our method.

### 2.5. Protein Profile of Neural Differentiation in GMSCs

In order to explore the effects of chitosan hydrogel supplemented with metformin and growth factors on the neural differentiation of stem cells, proteomics was used to construct the protein profiles to analyze the fluctuations of protein abundance after neural differentiation in GMSCs. The results showed that a total of 2230 proteins were detected by label–free proteomics (Figure 6a). The results of the difference analysis showed that, compared with GMSCs, a total of 213 differential abundance proteins (DAPs) were found and identified in nerve–like cells under 3D hydrogel conditions (Figure 6b), of which 181 were upregulated (Appendix A) and 32 were downregulated (Appendix A). To understand the social relationship of DAPs, we analyzed the protein–protein interaction network. It was found that there is a complex interaction network relationship in DAPs (Figure 6c), and a high–scoring sub–network was found, including ATP5F1, ATP5J, NDUFS3, and GLUD1 (Figure 6d). Protein–protein interactions were performed via Cytoscape. All subnetworks and scores are available in Appendix A.

To interpret the biological functions and pathways of DAPs, we performed GO and KEGG analyses (Figure 7). The results showed that DAPs were involved in important biological processes, including a small molecule metabolic process, an oxidation–reduction process, cellular respiration, a single–organism metabolic process, energy derivation by oxidation of organic compounds (*p* < 0.001), an organic acid metabolic process, the generation of precursor metabolites and energy, a carboxylic acid metabolic process, a nucleoside Triphosphate metabolic process, and an oxoacid metabolic process. These molecules are mainly located in mitochondrial part, mitochondrial matrix, mitochondrion, cytoplasmic part, organelle inner membrane, mitochondrial inner membrane, intracellular organelle part, organelle part, cytoplasm, and mitochondrial protein complex. It was found that the main molecular functions of these proteins include oxidoreductase activity, catalytic activity, hydrogen ion transmembrane transporter activity, electron carrier activity, oxidoreductase activity, acting on the aldehyde or oxo group of donors, NAD or NADP as a acceptor, cofactor binding, oxidoreductase activity, acting on the aldehyde or oxo group of donors, coenzyme binding, chaperone binding, and unfolded protein binding. A total of 28 pathways were discovered here, with *p* < 0.001. It was found that a subset of high scores, including for ATP5F1, ATP5J, NDUFS3, and GLUD1, was involved in neural nucleus development. In summary, the results of proteomics showed that neural differentiation involved complex protein interactions, and metformin upregulates multiple proteins that promote neurogenesis, including ATP5F1, ATP5J, NDUFS3, and GLUD1 (Figure 6d).

## 3. Discussion

This study emphasizes the regulatory effect of metformin on the neural differentiation of GMSCs in a 3D culture environment. In addition, the effect of metformin on GMSCs is illustrated, especially in maintaining pluripotency. Dental mesenchymal stem cells originate from the cranial neural crest [27]. Therefore, dental–derived mesenchymal stem cells are considered to have stronger neural differentiation potential than other mesenchymal stem cells, and dental–derived mesenchymal stem cells are also commonly used in the research of neurodegenerative disease [28]. Among them, GMSCs are derived from gingival tissue and expresses markers of neural crest, such as NESTIN, SNAI1, TWIST1, PAX3, SOX9, and FOXD3 [29], and can generate neurospheres. In the in vitro neurogenesis induction experiment, GMSCs can be induced into functional neurons to a certain extent, and GMSCs perform better than another dental mesenchymal stem cells and dental pulp stem cells (DPSCs) [30]. We used GMSCs as the neuroblastic–induced cells. The research on drug–induced neural differentiation can be roughly divided into several categories. Among them, Woodbury et al. pioneered the use of antioxidants, such as DMSO and BHA, and other chemical induction [31]. This method has high induction efficiency by changing the cytoskeleton, but it also has high cytotoxicity. In addition, there is research to change the differentiation fate of cells through gene editing [32]. In this study, we chose to induce combined growth factors and metformin. Among them, EGF and bFGF are commonly used in the cultivation of neural stem cells in vitro, which can promote cell proliferation and morphogenesis in vitro, and they are widely present in the central nervous system with neuroprotective effects [33]. VEGF is another important protein growth factor, as it can directly regulate the ion channels on the cell membrane of neurons, regulate the development and regeneration of neurons, and exert neuroprotective effects [34]. Brain–derived neurotrophic factors are mainly active in the hippocampus and cerebral cortex, and they play a neuroprotective effect by binding to receptors and promoting the proliferation, differentiation, and migration of NSCs [35]. In the work of Miller’s research team, it was found that metformin can stimulate the aPKC–CBP signaling pathway and promote the regeneration and development of neurons in the mouse brain [21]. In addition, metformin has been shown to restore the response of aging neural stem cells to pro–differentiation signals and promote the regeneration of nerve myelin. Therefore, we believe that metformin has the effect of promoting neural differentiation, and it can be used in combination with growth factors to promote the neural differentiation of GMSCs.

In this study, we verified the effect of metformin on the differentiation of GMSCs induced by osteogenesis, adipogenesis, and chondrogenesis. After 21 days of induction, they were stained with Alizarin red, Oil Red O, and Alcian blue; the additional metformin had no effect on multidirectional differentiation, but it can still maintain the potential for multi–lineage differentiation. There is evidence that metformin can inhibit stem–cell aging through the AMPK pathway [36]; moreover, it can promote osteogenic differentiation of stem cells [37].

The influence of GMSCs mesenchymal characteristics has also been tested. We selected the positive markers of mesenchymal stem cells, CD73, CD90, and CD105, which were tested by flow cytometry. In GMSCs culturing medium with and without metformin, positive expression was detected in most of the cells. Most of the negative markers of mesenchymal stem cells, such as CD11b, CD19, CD34, and CD35, are not expressed in most cells. Therefore, metformin can maintain the mesenchymal properties of GMSCs. The maintenance of mesenchymal stemness is closely related to the potential of cells to differentiate into specific tissues. Therefore, it is particularly important not to destroy the stemness of MSCs in the process of drug induction.

Recently, people are more and more enthusiastic about combining stem–cell therapy with 3D culture. The biocompatibility of hydrogels has shown its unique advantages in stem cell culture and sustained drug release [38]. Chitosan hydrogel is a naturally derived polymer. Because of its low biotoxicity and biodegradability, it is often used as a material for neural tissue engineering. The methods of hydrogel crosslinking include physical crosslinking and chemical crosslinking. The physical crosslinking method has low toxicity to cells and is suitable for cell culture [39].

We produced a temperature–sensitive chitosan hydrogel with a pore size of 20–40 μm, which has channels for transporting oxygen and macromolecular nutrients, which can provide the exchange of GMSCs in the hydrogel, and GMSCs can adhere, grow, and proliferate in the hydrogel. Previous studies have shown that chitosan plays an important role in nerve–tissue–regeneration engineering [40]. We compared the advantages of hydrogel culture for neurogenic induction, and, under the conditions of hydrogel culture, we compared the advantages of adding metformin and without metformin. The results suggest that hydrogel culture can harvest a small number of TUJ1 positive cells and NES positive cells, of which the positive expression of *TUBB3* is not significant, and the addition of inducing drugs based on 3D culture can harvest a large number of TUJ1 positive cells. Compared with the control group, NES–positive cells from the Met–CS group have a significant increase; moreover, they also have a significant increase compared with the hydrogel culture group. On the one hand, the structure of the hydrogel has an impact on the biomechanical properties of the cells and changes their development. On the other hand, the hydrogel contains the induced drug, and the multi–channel delivery of the drug makes the interaction with cells more frequent. Finally, the cell grows and proliferates in the hydrogel, and the interaction between the cells and the cell communication increase. Hydrogel simulates the “vivo–like” microenvironment, so cells can better secrete paracrine factors; the paracrine effect of cells directly affects cell development [18]. In addition, the specific changes of TUJ1 protein related to the cytoskeleton in the Met–CS culture system are due to the unique biomechanical effects of the 3D culture system. The insignificant changes in GFAP and MAP–2 indicate that the cells have not differentiated to mature neurons. In the study of Linares et al. [30], the induction of functional neurons may require neurosphere–mediated processes. The recently discovered combination of small molecules seems to enhance the neurogenic induction medium, skipping neurosphere mediation, and directly transforming fibroblasts into functional neurons [41].

This study describes and analyzes the protein profile of GMSCs induced by metformin in a 3D culture system. The study of high–throughput protein expression profiles can effectively explore the mechanism of metformin on the neural differentiation of GMSCs. To the best of our knowledge, our study uses quantitative proteomics for the first time to explore the effect of metformin on GMSCs. Metformin upregulates a series of proteins involved in many biological processes. Through the high score subset containing ATP5F1, ATP5J, NDUFS3, and GLUD1, we have determined that metformin promotes the neural differentiation of GMSCs by activating the neural nucleus development process and is mainly involved in the biological process of neural nucleus development.

As for the specific expression of the PAX6 gene in the 3D culture system, it may be related to the different degrees of biomechanical changes, resulting in different cell subpopulations. The neurodifferentiation promotion effect of metformin in chitosan hydrogel allows us to see the possibility of treatment of neurological injury diseases. Chitosan hydrogel can be degraded and absorbed in the body. It is often used as a carrier for sustained–release drugs in vivo. It can be used as a carrier for the transplantation of stem cells. Orthotopic transplantation after hydrogel degradation and the paracrine function of cells can treat neurological injuries and diseases in both ways. Furthermore, chitosan hydrogels can be surface–modified with nerve–inducing drugs, which can also maintain the nerve–inducing function in the body and stimulate the differentiation potential of GMSCs. The next research will explore this. In addition, the expression of early neural markers allowed us to see the dawn of the hydrogel–inducing mature neurons in the cultivation potential of neurospheres. Furthermore, the use of hydrogel as a cell carrier may be used as a method of administration for nerve injury or neurodegenerative diseases, which may include the differentiation process in vitro and the sustained–release process of cytokines in vivo. We realize that this study is limited. On the one hand, neural–like cell differentiation is still preliminary, and subsequent work should be performed to construct mature neuronal differentiation systems and evaluate their applications in neuroregenerative medicine. On the other hand, although we elucidate the mechanism of neural differentiation through molecular biological means, we do not understand the key genes, proteins, or ligands that regulate neural differentiation, suggesting that further work should be carried out to extend the precise regulation of neural differentiation.

## 4. Materials and Methods

### 4.1. Materials

Dulbecco Modified Eagle Medium (DMEM, Biological Industries, Kibbutz Beit Haemek, Israel), fetal bovine serum (FBS, Biological Industries, Kibbutz Beit Haemek, Israel), 1× penicillin–streptomycin (Biological Industries, Kibbutz Beit Haemek, Israel), EGF, bFGF, BDNF, VEGF (all from Sino Biological, Beijing, China), Metformin (Sigma, MO, USA), PBS (Biological Industries, Kibbutz Beit Haemek, Israel), Trypsin (Biological Industries, Kibbutz Beit Haemek, Israel), chitosan (Macklin, Shanghai, China), β–glycerophosphate (Macklin, Shanghai, China), Nestin Rabbit Monoclonal Antibody, GFAP Rabbit Monoclonal Antibody, TUJ1 Rabbit Monoclonal Antibody(all from Beyotime, Shanghai, China), FITC–labeled Goat Anti–Rabbit IgG(H+L), Cy3–labeled Goat Anti–Rabbit IgG (H+L) (all from Beyotime, Shanghai, China), HRP–labeled Goat Anti–Rabbit IgG (H+L) (Beyotime, Shanghai, China), and 4% Paraformaldehyde (Service, Wuhan, China).

### 4.2. Synthesis of CS/β–GP Hydrogel

The chitosan hydrogel was prepared as previously reported [39]. Simply, a stock solution of 2% chitosan was prepared by dissolving the chitosan powder (Macklin, Beijing, China) in 0.1 M acetic acid, followed by a filter–sterilization, and storage in an ice bath. The 50% β–glycerophosphate solution was prepared by dissolving β–glycerophosphate powder in water, followed by sterilization with 0.22 μm filters. The β–glycerophosphate solution was added to the chitosan solution at a 1:9 volume, under stirring, inside the ice bath. The gelation time of the chitosan/β–glycerophosphate (CS/β–GP) hydrogels was different depending on the volume ratios between chitosan and β–glycerophosphate. Finally, the two solutions were fully mixed.

### 4.3. Characterization of CS/β–GP Hydrogel

#### 4.3.1. Observation by SEM

Prepared solutions of the chitosan and CS/β–GP were placed into the wells of a 12–well plate, at a volume of 500 μL/well. After gelation, the chitosan hydrogel was freeze–dried under vacuum for 36 h following being cross–sectioned. After sputter–coating with gold, the hydrogel was observed under the SEM (SEM, S4800, Hitachi Ltd., Tokyo, Japan). The mesh size was measured by using ImageJ software (version 1.8.0).

#### 4.3.2. FTIR Analysis

Fourier–transform infrared (FTIR) was used to characterize the chemical properties of chitosan hydrogel. FTIR spectroscopy was performed by using an FTIR spectrometer (Bruker Equinox 55 FTIR, Karlsruhe, Germany).

### 4.4. Cell Culture and Treatment

The human gingival mesenchymal stem cells were donated by Kangyanbao Stem Cell Technology Co., Ltd. GMSCs were cultured in fresh medium, which consisted of Dulbecco’s minimum essential medium (DMEM, BI, Kibbutz Beit Haemek, Israel), 10% fetal bovine serum (FBS, BI, Kibbutz Beit Haemek, Israel), 1% 100 U/mL penicillin, 100 mg/mL streptomycin (BI, Kibbutz Beit Haemek, Israel), and 100 μmol L–ascorbic acid (BI, Kibbutz Beit Haemek, Israel). All experiments used the Passage 3–5 of GMSCs. The culture medium was altered every 2–3 days and passaged when the cell fusion rate reaches 80%. The GFP–GMSCs cells used in the experiment were constructed from our laboratory, and the GFP protein was inserted into the GMSCs coding sequence through a lentiviral vector, and the GFP protein was stably expressed. In the drug induction experiment, the optimal concentration of metformin is 100 mM [42], and the concentration of growth factor is 10 ng/mL according to previous studies [43].

### 4.5. Cell Viability Assay on the Hydrogel

We mixed the GFP–GMSCs resuspended in PBS with the CS/β–GP solution in different volume ratios in an ice–water bath. The volume ratios were 1:3, 1:5, and 1:9. The total hydrogel culture system was 200 μL. A total of 1 × 10^5^ cells were inoculated into a 24–well plate, and then we added 2 mL of the complete medium after the hydrogel was solidified. We randomly took 5 pictures for each well at 0, 4, 8, 16, and 24 h with a fluorescence microscope, and we used ImageJ to semi–quantitatively analyze the fluorescence images through a fluorescence microscope (EVOS FL Auto, Olympus, Tokyo, Japan).

### 4.6. Characterization of Adult Human GMSCs

The immunophenotype of GMSCs and metformin–treated GMSCs was detected by using flow cytometry analysis. The GMSCs were centrifuged in EP tubes at a concentration of 1 × 10^6^ cells/mL. We removed the supernatant, washed with PBS, and then incubated with paraformaldehyde. The surface biomarkers of GMSCs were bound by CD34–PE (BD Biosciences, NJ, USA), CD11b–FITC (BD Biosciences, NJ, USA), CD19–PC5.5 (BD Biosciences, NJ, USA), CD45–APC (BD Biosciences, NJ, USA), HLA–DR–PE (BD Biosciences, NJ, USA), CD73–FITC (BD Biosciences, NJ, USA), CD90–PC5.5 (BD Biosciences, NJ, USA) and CD105–APC (BD Biosciences, NJ, USA). After labeling and counting, the characteristics of fluorescence were analyzed and recorded.

### 4.7. Multi–Lineage Differentiation of GMSCs

To verify the multi–lineage differentiation potential of GMSCs and metformin–treated GMSCs, we added metformin to the osteogenic, adipogenic, and chondrogenic induction medium (all from Cyagen, CA, USA), respectively, and the control group was a conditioned induction medium without metformin.

For osteogenic differentiation, the uncharacterized cells were seeded in 12–well plates (Corning, NY, USA) at a density of 5 × 10^4^ cells/well and cultured in the proliferating medium until reaching 70% confluence. For osteogenic differentiation, GMSCs were replaced with DMEM supplemented with 10% FBS, 0.1 mM dexamethasone, 10 mM β–glycerophosphate, and 50 mM ascorbic acid for 4 weeks. The medium was changed every 3 days. The calcification of an extracellular matrix was assayed by Alizarin red (Cyagen, CA, USA) staining and captured by the Compact Cell Culture Microscope, CKX3 (Olympus).

For chondrogenic differentiation, GMSCs were induced by the chondrocyte differentiation basal medium for 3 weeks and then assayed by 1% Alcian blue (Cyagen, CA, USA) staining to detect the synthesis of proteoglycans by chondrocytes.

For adipogenic differentiation, GMSCs were induced by the adipogenic differentiation basal medium for 3 weeks and then assayed by Oil Red O (Cyagen, CA, USA) staining to detect the formation of lipid droplets by adipocyte.

### 4.8. Differentiation of GMSCs into Neuronal Cells on the Hydrogel

In the process of preparing chitosan hydrogel, we dissolved β–glycerophosphate sodium in water containing the growth factor, and we stirred and mixed it with the chitosan solution in an ice–water bath at a volume ratio of 1:9. The prepared chitosan hydrogel was stored in an ice–water bath, and the final growth factor concentration was 10 ng/mL. We then took the GMSCs cultured in the T25 culture flask, discarded the medium, washed twice with PBS, digested with trypsin, stopped the digestion with complete medium, centrifuged at 1000 rpm for 5 min, resuspended the cells in PBS, and mixed with hydrogel with a volume ratio of 1:3 on an ice–water bath. Then we inoculated them into a 24–well plate. We placed them in a 37 °C incubator, and they solidified after a few minutes. In order to maximize the neurological differentiation of GMSCs, we used the method of segmented induction. In the first 5 days of culture, we added complete medium to the hydrogel, and then we added complete medium containing metformin for the next five days, called the Met group (Figure 2.) The hydrogel group refers to a hydrogel culture system that does not add metformin and is inoculated with the same amount of cells. The blank group refers to a complete medium without growth factors, metformin, and hydrogel. The inoculum amount and induction time were the same as those of the other two groups, both were 14 days, and the medium was changed every 2 days. The GMSCs in a 3D cellular culture hydrogel containing metformin and growth factor are termed the Met–CS groups and the GMSCs cultured in a hydrogel without metformin and growth factors are termed the CS groups.

### 4.9. Quantitative Real–Time Reverse–Transcription Polymerase Chain Reaction (qRT–PCR)

The quantitative gene expression of the neural–lineage markers was assessed by real–time PCR (ABI StepOne, CA, USA). Total RNA was extracted, using the Trizol method (CWBio, Beijing, China), and converted to complementary DNA (cDNA) by the TaqMan Reverse Transcription Kit (CWBio, Beijing, China) containing random hexamer primer mix. RNA was quantified and preserved. The gene template was added to the real–time fluorescence quantitative amplification system, and all primers were listed in Table 2. All reactions were carried out in triplicate. All of the mentioned gene Ct values of each parameter were normalized by using the reference gene Ct value to define the Ct value based on the UltraSYBR Mixture (CWBio, Beijing, China). GAPDH was used as an internal control.

### 4.10. Western Blot

The total protein was extracted and redistributed by 10% SDS–PAGE after being mixed with 5× loading buffer. The mixture was separated by using an electrophoresis program, according to molecular weight, and transferred to 0.22 μm PVDF (Millipore, MA, USA) membrane by JY–ZY5 transfer electrophoresis tank (Junyi, Beijing, China) at 150 mA for 1.5 h. After 1 h of blocking with 5% skimmed milk (BD Biosciences, NJ, USA), the protein of the stem cells was bound by Tuj1 antibody (Beyotime, Shanghai, China), GFAP antibody (Beyotime, Shanghai, China), and Nestin antibody (Beyotime, Shanghai, China) at 4 °C for 16–20 h. The secondary antibody with HRP (Proteintech, IL, USA) was used to incubate at 37 °C for 1 h, and the protein abundance was observed after mixing with ECL luminescent solution (Beyotime, Shanghai, China).

### 4.11. Immunofluorescence

After a 10–day culture, the morphological change was investigated for differentiation by immunofluorescent staining. The GMSCs were validated by the neuronal genes profiling with NES, TUJ1. The samples were fixed with paraformaldehyde for 30 min and then incubated with primary antibody against NES and TUJ1 (Beyotime, Shanghai, China) at 4 °C, overnight. Then the cells were incubated with appropriate fluorescently labeled Cyc3–labeled goat anti–rabbit IgG and FITC–labeled goat anti–rabbit IgG (Beyotime, Shanghai, China). The mounting solution contained DAPI (Beyotime, Shanghai, China) and was used to mount the cells, which were then observed under a confocal laser microscope (Leica, TCS SP8 SR).

### 4.12. Proteomics and Bioinformatics Analysis

GMSCs were added with 8M urea (Xilong Scientific, China) in 100 mM TEAB (Thermo Fisher Scientific, MA, USA) and lysed in ice for 30 min. After centrifugation at 12,000 rpm for 10 min, the supernatant was transferred and stored. A total of 300 μg of protein was extracted and mixed with DL–dithiothreitol (DTT, Macklin, Beijing, China) and Iodoacetic acid (IAA, Macklin, China) in trypsin at 37 °C, overnight, to prepare polypeptides. The mixture was desalted at HLB 1cc Extraction Cartridges (Waters, MA, USA). After drying, the peptide was re–dissolved in 1% formic acid (FA, Rhawn, Beijing, China) and detected by mass spectrometry. The mobile phases carrying polypeptides were redistributed in the C18–reversed–phase (Gemini columns, CA, USA). The eluent was identified based on Orbitrap Fusion MS (Thermo Fisher Scientific, MA, USA) equipped with an online Easy–NLC 1000 system (Thermo Fisher Scientific, MA, USA). The original file from MS/MS was read in Maxquant software (version 1.6.2.0) based on the human FASTA database of Uniprot (17 February 2022, https://www.uniprot.org/). The false discovery rate (FDR) was limited to 0.01 by using the integrated tool. Gene ontology (GO) analysis and Kyoto Encyclopedia of Genes and Genomes (KEGG) were enriched and clustered by DAVID tool (17 February 2022, https://string-db.org/cgi/input.pl). The protein–protein interaction network was constructed by STRING (17 February 2022, https://string-db.org/cgi/input.pl), which was visualized by Cytoscape (version 3.7.2) and extracted the subnetwork by the plug–in MCODE.

### 4.13. Statistical Analysis

All experiments were conducted three times to ensure reproducibility. Fluorescence image statistics used ImageJ software (version 1.8.0), and data statistics were performed through GraphPad Prism (version 8.0). The data were expressed as the mean ± standard error of the mean (SEM); the difference between the experimental groups and the control group were compared by using Tukey’s Multiple Comparison Test via GraphPad Prism (San Diego, CA, USA). Significance was calculated by using a *t*–test: *p* < 0.05 *, *p* < 0.01 **, and *p* < 0.001 ***. Meanwhile, fold change (FC) was required to |FC| ≥ 1.5.

## 5. Conclusions

Therefore, these findings indicated the potential of using chitosan hydrogel supplemented with metformin to induce neurogenic differentiation of GMSCs in vitro for the further treatment of neurological injury diseases.

## Data Availability

Data are available via ProteomeXchange with identifier PXD030306.

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
