# Peer review of "Chitosan Hydrogel Supplemented with Metformin Promotes Neuron–like Cell Differentiation of Gingival Mesenchymal Stem Cells"

_ijms, 2022, doi:10.3390/ijms23063276_

Round 1
Reviewer 1 Report
The authors of this study investigated the effect of metformin on multi-lineage differentiation potential of human gingival mesenchymal stem cells (GMSCs) in a 3D culture environment. They used quantitative proteomics to explore the effect of metformin on the neural differentiation of GMSCs . Their results show that chitosan hydrogels containing metformin promote the up-regulation of numerous neural regeneration-related proteins but that the cells did not differentiated to mature neurons.
Main Comment:
This study presents a huge amount of data, but these data are not sufficiently discussed. It would be interesting to know, for example, if identified up-regulated proteins could support true neuronal behavior, i.e. electrical activity of differentiated GMSCs.
Specififc Comments:
There are abbreviations within the text that are explained in Supplementary Tables but not in the text. According to my opinion they should be explained also in the main text when used for the first time.
Abstract: there are several abbreviations that are not explained (NES, TUJ1, ATP5F1, ATP5J, NDU-FS3, GLUD1)
Line 48-49: ” The ability to differentiate into nerve-like cells and the potential for nerve regeneration from the tooth and oral stem cells has been demonstrated in vivo“ - Reference is missing.
Lines 63-68: abbreviations bFGF, BHA, GMSCs, TCPS are not explained
Line 71: EGF, bFGF abbreviations are not explained
Line 108: Where is „ImageJ“?
Line 117: „2.2....“ missplaced text
Line121: What is SEM image?
Line 125: “To verify the pluripotency ....“ This sentence is not complete
Line 129: intracellular
Line 168: should be apparently (Fig. 3a, Fig. 3c)
Line 199: abbreviations EGF, bFGF, BDNF, and VEGF are not explained
Line 233-234: This sentence should be rewritten
Line 260-261: ATP5F1, ATP5J, NDUFS3, GLUD1 abbreviations are not explained
Line 290: a subset
Line 305: „ OSCs originate...“ wrong or missplaced sentence
Line 313: What is “DPSC“ and why it is no mentioned in the Inroduction?
Line 317: abbreviation BHA is not explained
Line 389-392: „Through a high...“ It is not clear what the authors wanted to say. This sentence should be rewritten
Figure 7: It seems that this figure is mentioned nowhere in the text and is not discussed. It shows interesting interpretation of quantitative proteomics data and the authors should consider the possibility to include a specific paragraph about this analysis into the Discussion.
Reviewer 2 Report
The manuscript gives a brief explanation on the use of metformin in the 3D- culture system of chitosan/Glycerophosphate for gingivial mesenchymal stem cells (GMSC). The authors have tried to show that the induction of the GMSC in the chitosan hydrogel with metformin can elicit the differentiation of the GMSC to neurons. The manuscript is in general well written, but it would need some addition and some correction in experimental data from the authors to explain the potential readers on the use of metformin in such system. Such additions and corrections are summarized below in the minor comments.
Minor comments:
- Authors have tried to show that the GMSC can be differentiated into different lineages of osteogenic, adipogenic and chrondrogenic cells with various experiments like oil-red (Adipocytes) staining, alizarin red staining (osteoblasts) and alcin staining (for chondrocytes). However, in none of these experiments, the authors have tried to explain how efficient was the differentiation protocol and what percentage of cells were actually differentiated in such cells from the total cellular population and authors should give the readers some highlight on same.
- The quality of blot for proteins NES and GFAP is poor in figure 5b and please replace these blots with a more clear blot. Also, please add blot for SOX1 in the same figure and the same will also justify RNA expression as checked in the above figure.
- As mentioned on the line 260 a high score for the sub-network was found after proteomic analysis. Please cite the source where this analysis was done and also provide the score with categorization for scores.
- In the proteomics analysis on page 9, authors don’t need to give p-score for every analysis, they can just mention once that the proteomic analysis with p<0.001 and then continue from their describing different pathways.
- In the whole discussion section, the authors have refrained from discussing the disadvantage of this system with use of metformin in Chitosan hydrogel. I encourage authors to please provide the readers also about some disadvantage from the use of metformin in chitosan hydrogel scaffold, especially while doing experiments. As this would provide potential users of this system in careful with the use of metformin in chitosan hydrogel.
